# Endoscopic Ultrasound-Guided Fine Needle Biopsy of Focal Liver Lesions: An Effective Mini-Invasive Alternative to the Percutaneous Approach

**DOI:** 10.3390/diagnostics14131336

**Published:** 2024-06-24

**Authors:** Gabriele Rancatore, Dario Ligresti, Giacomo Emanuele Maria Rizzo, Lucio Carrozza, Mario Traina, Ilaria Tarantino

**Affiliations:** 1Endoscopy Service, Department of Diagnostic and Therapeutic Services, IRCCS-ISMETT, 90127 Palermo, Italy; gabriele.rancatore@gmail.com (G.R.); lcarrozza@ismett.edu (L.C.); mtraina@ismett.edu (M.T.); itarantino@ismett.edu (I.T.); 2Section of Gastroenterology & Hepatology, Department of Health Promotion, Mother and Child Care, Internal Medicine and Medical Specialties, PROMISE, University of Palermo, 90121 Palermo, Italy

**Keywords:** EUS, liver, tumor, lesion, FNB

## Abstract

Despite the introduction of serological neoplastic biomarkers and typical radiological characteristics in clinical practice, liver biopsy (LB) is often still necessary to establish a histological diagnosis, especially in ambiguous cases. Nowadays, LB via the percutaneous approach (PC-LB), under computed tomography (CT) scan or ultrasonography (US) guidance, is the route of choice. However, certain focal liver lesions can be challenging to access percutaneously. In such cases, endoscopic ultrasound (EUS)-guided fine needle biopsy (FNB) may represent an attractive, minimally invasive alternative. This retrospective observational study aimed to evaluate the efficacy, diagnostic performance, and safety of EUS-FNB conducted on 58 focal liver lesions located in both liver lobes. The adequacy of FNB samples for focal liver lesions located in the left and right lobes was 100% and 81.2%, respectively, and the difference was statistically significant (*p* = 0.001). Technical success was 100% for both liver lobes. The overall sensitivity and specificity were 95% and 100%, respectively. EUS-FNB is effective in making an accurate diagnosis with an excellent safety profile for focal liver lesions located in both liver lobes.

## 1. Introduction

Despite the introduction of serological neoplastic biomarkers and typical radiological characteristics useful for the differential diagnosis of focal liver lesions in clinical practice, liver biopsy (LB) is often still necessary to make a histological diagnosis, especially in ambiguous cases [1]. The percutaneous approach (PC-LB), guided by computed tomography (CT) or ultrasonography (US), is the standard route of choice. However, PC-LB via ultrasound or CT guidance can be challenging for many reasons, such as for anatomical reasons like focal liver lesions located in the left liver lobe, near the gallbladder, or next to big vessels. Moreover, it is a type of biopsy not often performed under deep sedation, so difficulties in sampling due to patient movement can arise. Similarly to ultrasound, but unlike CT, EUS-guided biopsy of focal liver lesions is controlled “in real time”, reducing the risk of bleeding, pneumothorax, and gallbladder perforation. An alternative approach, in cases of difficult localization or when the percutaneous route is not feasible, is the transjugular route (TJ-LB). Although some of the limitations of the percutaneous approach can be overcome with TJ-LB, this latter method is a more complex procedure, with potential complications including injury to the neck vessels resulting in bleeding, hematoma and arteriovenous fistula formation [2]. In recent years, endoscopic ultrasound (EUS) has emerged as an important tool in the detection of focal liver lesions less than 1 cm, outperforming US, CT, and MRI [3,4,5]. Furthermore, this ability increases with the use of contrast-enhanced harmonic EUS (CEH-EUS) [6]. The first study that described the diagnostic yield of EUS-guided focal liver lesion tissue acquisition was conducted in 1999 by Nguyen et al. In this prospective study, EUS-guided fine needle aspiration (EUS-FNA) of focal liver lesions was capable of establishing a cytohistological diagnosis in all patients, and could detect small liver lesions that were not visualized under CT [7]. Interestingly, an international investigation of 167 cases of EUS-FNA of focal liver lesions demonstrated that EUS-FNA could obtain a diagnosis when previous US- or CT-guided FNAs were unable to do so [8]. For these reasons, it is currently believed that EUS is at least as effective as traditional methods in performing liver biopsy [9]. EUS-LB has several advantages, such as the capacity to access focal target lesions whilst avoiding adjacent structures and blood vessels, as well as providing a real-time detailed view of the biopsy needle route during tissue acquisition through the liver, minimizing the rate of adverse events [5,10,11,12,13]. Furthermore, EUS-LB is performed under deep sedation, guaranteeing reduced procedural anxiety and increased compliance [14,15]. Finally, the possibility to integrate real-time elastography (RTE) and contrast enhancement (CE) improves the diagnostic performance in focal lesions [16] by providing semi-quantitative measurements of lesion stiffness and vascular behavior [17,18]. Due to the close proximity of the probe to the liver surface, EUS is also advantageous in patients with massive ascites, which represent a relative contraindication for the percutaneous approach due to the increased risk of bleeding. These advantages, combined with a low rate of adverse events, make EUS an excellent modality for the diagnosis and staging of focal liver lesions. Despite these benefits, it is still considered an alternative diagnostic method, used only after unsuccessful or infeasible attempts through the traditional routes [19,20,21]. Many of the studies in the literature were conducted with FNA needles, which were found to be inferior to FNB needles in a prospective study by Gheorghiu et al. with a statistically significant difference (100% vs. 86.7%, *p* = 0.039) [22]. FNBs are also suitable for molecular analyses, as demonstrated by Choi et al., where the next-generation sequencing (NGS) method improved the overall diagnostic accuracy of EUS-guided tissue sampling [23]. However, in all of the studies mentioned above, the lesions were located only in the left liver lobe. Only one previous study, conducted by Chon et al., evaluated the efficacy and safety of EUS-FNB of focal liver lesions located in both liver lobes, but they did not report differences in diagnostic yield between lesions located in the right or left lobes [20]. The objective of our study was to assess the diagnostic yield and safety of EUS-LB in patients with focal liver lesions in both liver lobes using only FNB needles, and to report any possible differences in the diagnostic yield between lobes.

## 2. Materials and Methods

All patients who underwent EUS-FNB of focal liver lesions at our institute between 2015 and 2023 were retrospectively analyzed. The indication to perform EUS-LB was the presence of a focal liver lesion poorly accessible via the percutaneous approach or incidentally found during an EUS examination performed for other reasons, and whose histopathological result could impact case management, according to the European Society of Gastrointestinal Endoscopy (ESGE) guidelines [24]. We included all adult patients with focal liver lesions, regardless of underlying advanced chronic liver disease. Exclusion criteria included contraindications for deep sedation and bleeding disorders, the latter identified by an international normalized ratio >1.5 or platelet count <50,000/mm^3^. According to the British Society of Gastroenterology (BSG) and ESGE guidelines, oral anticoagulants were temporarily replaced with low-molecular-weight heparin, P2Y12 receptor antagonists were discontinued five days before the procedure, and aspirin was continued [25].

### 2.1. Diagnostic Performance

Our study aimed to evaluate the diagnostic performance and safety of EUS-FNB of focal liver lesions. We assessed technical success in terms of adequacy and accuracy. Adequacy was defined as the ability of the method to establish a histopathological diagnosis. Accuracy was defined as the concordance between the definitive histology on the surgical specimen (for resectable patients), radiological characteristics, and clinical course during follow-up. Sensitivity and specificity were also evaluated. Sensitivity was defined as the probability that malignant focal liver lesions would give a positive test result. Specificity was defined as the probability that benign focal liver lesions would give a negative test result. Positive predictive value (PPV) corresponded to the proportion of biopsies testing positive for malignancy that were correctly diagnosed as malignant. Negative predictive value (NPV) corresponded to the proportion of biopsies testing negative for malignancy that were correctly diagnosed as benign. Malignancy was defined as “a case with histologically malignant findings”, and benign cases were those with “histologically benign findings on EUS-FNB and no tumor growth even after one-year follow-up”. Adverse events (AE) were defined according to the AGREE classification [26].

### 2.2. Endoscopic Ultrasound-Guided Fine Needle Biopsy Procedures

All procedures were performed with patients under deep sedation using propofol with the assistance of an anesthesiologist and a nurse specialized in sedation. No antibiotic prophylaxis was administered. EUS-FNBs were performed by expert endosonographers using a linear array echoendoscope (Olympus GF-UCT180 and Olympus GF-UCT140P with ultrasound processor EU-ME2, Olympus Corporation, Tokyo, Japan). Tissue acquisition was performed with 19-, 20- or 22-gauge FNB needles (ProCore, Cook Medical Inc., Winston-Salem, NC, USA; Acquire, Boston Scientific, Marlborouh, MA, USA) using the wet suction technique. The choice of gauge was per operator preference and based on personal familiarity with a type of needle. The target liver lesions were identified in B-mode and better characterized with real-time Doppler. After using real-time Doppler to avoid blood vessel puncture, EUS-FNB was performed (Figure 1). FNB was performed from the stomach for lesions located in the left liver lobe and from the duodenal bulb for right lobe lesions. Before puncturing the lesion, the stylet was removed, the needle was flushed with saline to replace the column of air, and a pre-vacuum 20 mL syringe was utilized to apply suction. Once the lesion was punctured, the sample was collected by performing a back-and-forth movement of the needle within the lesion, applying negative pressure using a 20 mL syringe. Afterwards, aspiration was stopped by closing the syringe latch and the needle was removed. The samples were placed in a formalin bottle by reinserting the stylet. At the end of the procedure, the patients were monitored for 24 h and discharged if no AEs occurred. In the case of AE onset, patients were admitted for further investigation.

### 2.3. Histological Evaluation

The obtained fragments were placed in formalin and inspected to evaluate whether further needle passes were necessary according to the VOSE (Visual On-Site Evaluation) technique [27]. A tissue sample was defined as adequate when judged quantitatively sufficient to establish a histopathological diagnosis by expert pathologists. The specimens obtained after being fixed in formalin and embedded in paraffin were evaluated using hematoxylin–eosin staining. If the diagnosis was not established with hematoxylin–eosin staining alone, the expert pathologist applied immunohistochemistry based on clinical suspicion.

### 2.4. Statistical Analysis

Categorical variables were described as frequencies and percentages (%), while continuous variables were described as mean ± standard deviation (SD) or median with interquartile range (IQR) if not normally distributed. To investigate whether there was a statistically significant difference between the diagnostic accuracy of EUS-FNB of the left and right lobes, the Student’s *t*-test was performed. Inferential analysis for diagnostic yield was conducted using 95% confidence intervals (CI). The significance level was 0.05. Data processing and statistical analysis were performed using the following software: Microsoft Excel (version 2017, Microsoft Corporation, Redmond WA, USA), SPSS (version 20, SPSS Inc., Chicago, IL, USA) and Med Calc (Version 20, MedCalc Software Ltd., Ostend, Belgium).

## 3. Results

A total of 58 patients were included in our analysis. Of these, 32 patients (55%) were male, the median age was 61.5 years (SD: ±13.2), and the mean body mass index (BMI) was 28 Kg/m^2^ (Table 1). The localization of lesions was 72.4% (42 cases) in the left lobe and 27.6% (16 cases) in the right lobe. Only two patients had lesions in segments 7 and 8 of the liver, which are the furthest from the stomach and duodenum. The focal liver lesion localizations are summarized in Table 2. The mean tumor size on EUS was 34.3 mm (SD: ±21.5), and the median number of needle passes was two (IQR: 3–4). Of the 58 patients included in our study, 43 (74%) had a final diagnosis of malignancy, including 7 (15%) cases of hepatocellular carcinoma, 16 (38%) cases of cholangiocarcinoma, 16 (38%) metastases, 3 (7%) cases of malignant neoplasm of unclear origin, and 1 (2%) Kaposi sarcoma. A total of 12 patients (21%) received a final diagnosis of benign disease, and in all cases these were adenomas (Table 3). Only in three cases (5%) was the specimen suboptimal for histological evaluation, and all of these focal liver lesions were located in the right lobe (in segments 5, 6 and 7). The overall sensitivity and specificity were 95% and 100%, respectively, while overall adequacy and accuracy were 95% and 96%, respectively (Table 4). Subgroup analysis for different FNB needle gauges showed different diagnostic yields. In particular, adequacy and accuracy were 100% and 95% for 19-gauge FNB needles, both 100% for 20-gauge FNB needles, and 90% and 96% for 22-gauge FNB needles. Of note, all three inadequate biopsies were performed with a 22-gauge needle. On the other hand, subgroup analysis for different FNB needle types showed adequacy and accuracy of 94% and 95%, respectively, for Acquire needles (Boston Scientific, Natick, MA, USA), and adequacy and accuracy of 100% each for ProCore needles (Cook Medical Inc., Winston-Salem, NC, USA) (Table 5). Technical success was 100% for both right and left liver lobes. The adequacy of focal liver lesions located in left and right lobes was 100% and 81.2%, respectively, and the difference was statistically significant (*p* = 0.001). We also compared the diagnostic yield of EUS-LB between the right lobe and left lobe masses (Table 6). The median size of lesions on EUS and the median number of needle passes did not differ significantly between the left and right lobes. The sensitivity for lesions located in the left and right liver lobes was 96% and 91%, respectively (*p* = 0.001). Specificity was 100% for lesions located in both lobes. Positive predictive value (PPV) was 100% for lesions located in both liver lobes. Negative predictive values (NPVs) for lesions located in the left and right liver lobes were 90% and 50%, respectively (*p* = 0.001). No intra- or postprocedural adverse events were recorded according to the AGREE classification [26] (Table 7).

## 4. Discussion

Histological assessment through LB is often still necessary for diagnosis and staging of focal liver lesions, especially in ambiguous cases. PC-LB is a well-established procedure; it is simple, widely used, relatively safe and inexpensive. Unfortunately, there are anatomical reasons that can make PC-LB difficult to perform. EUS-LB has emerged as an attractive alternative tool in the diagnosis and classification of focal liver lesions [19]. EUS-LB has been shown to be a simple procedure with a good safety profile, as demonstrated in a meta-analysis conducted by Zeng et al., where it showed an average adverse event rate of only 3% [28]. Many studies have evaluated the diagnostic yield of EUS-guided focal liver lesion biopsy, reporting favorable results, with technical success rates of 86–100%, 88–100% sensitivity, 99–100% specificity, and an adverse event rate of 0–4% [8,29,30,31]. However, most studies have been based on the evaluation of FNA, which has been demonstrated to be inferior to FNB in terms of adequacy and accuracy by Gheorghiu et al. In fact, in this prospective study, the diagnostic accuracy of 22-gauge EUS-FNB needles (Franseen) was higher than that of 22-gauge EUS-FNA needles (100% vs. 86.7%, *p* = 0.039), with better sensitivity (100% vs. 83.3% *p* = 0.019) and no adverse events [22]. These results are in line with those from our retrospective analysis in terms of accuracy, sensitivity and safety, although, unlike at our institute, all specimens were collected using the “dry suction” technique. EUS-guided focal liver lesion biopsy has been shown to establish a diagnosis in cases where the percutaneous approach using either FNA [8] or FNB needles [32] has failed. In the latter study, conducted by Lee, Y.N. et al. [32] on 21 patients utilizing only FNB needles (22- and 25-gauge) with a mean needle pass of 2.1, the overall diagnostic accuracy for malignancy and specific tumor type were 90.5% and 85.7%, respectively, which are slightly lower than those observed in our study. In 2020, Chen et al. [33] evaluated the application of EUS-FNB in the diagnosis of the left-lobe HCC in cirrhotic patients with contraindications to percutaneous biopsy. In this study, where the application of suction was performed at the discretion of the endosonographers, the sensitivity, specificity, positive predictive value, negative predictive value, and accuracy rates of EUS-FNB in the diagnosis of HCC were 88.0% (22/25), 100.0% (5/5), 100.0% (22/22), 62.5% (5/8), and 90.0% (27/30), respectively [33]. Compared to our results, where no adverse events were observed, self-limiting bleeding occurred in three patients, probably reflecting the different characteristics of the included population who were affected by advanced chronic liver disease. On the contrary, in our study, only two patients were affected by advanced chronic liver disease. In our cohort, 5 (8%) focal liver lesions were located on the caudate lobe and were correctly sampled without adverse events. The caudate lobe is difficult to reach with the percutaneous approach because it is located deeply. On the contrary, the caudate lobe is clearly visible from the stomach under EUS guidance, so the target lesion is much closer than with the percutaneous approach, the distance to travel for the needle is shorter, and consequently the risk of vascular damage is reduced, making the procedure safer. Recently, Takano et al. assessed the feasibility of EUS-guided biopsy of focal liver lesions located in the caudate lobe. In this study, as in ours, 100% technical success and 100% adequacy were reported, demonstrating that the EUS approach can be the first choice for tissue acquisition in caudate lobe lesions [34]. All of the above-mentioned studies analyzing FNB needles were performed on focal liver lesions located in the left liver lobe. There are only a few previous studies that evaluated the diagnostic yield of EUS-guided focal liver lesion biopsy for lesions located in both liver lobes exclusively using FNB needles. A study published by Chon et al. evaluated the efficacy and safety of EUS-guided focal liver lesion biopsy located in both liver lobes with a mean number of needle passes of 2.6. In this retrospective analysis, the overall diagnostic accuracy and sensitivity were both 89.7%, specificity was 100%, and specimen adequacy for histology was 91.4% using 20-, 22- and 25-gauge FNB needles. Only one (out of 58 patients) grade III adverse event was registered. However, in the study conducted by Chon et al., differences in diagnostic yield between lesions located in the right or left liver lobes were not reported [20]. These results are slightly lower than ours. We believe that these differences are probably attributable to the different sampling techniques used in the latter study: the “dry suction “and “slow pull” techniques. In the literature, the superiority of the “wet suction” compared to the “slow pull” technique was found in EUS-LB. Specifically, in the multicenter study by Sharma et al., the “wet suction” technique produced larger histological specimens than the “slow pull” technique [35]. Furthermore, a study by Crinò et al. demonstrated that the tissue core percentage and tissue integrity score were slightly higher when using the “wet suction” technique [36]. However, it should be underlined that these are studies conducted on parenchymal liver disease. To the best of our knowledge, our study is the first that simultaneously evaluates the different diagnostic yield and safety of EUS-guided focal liver lesion biopsy for lesions located in both liver lobes using FNB needles only and the “wet suction” technique. The statistical differences in specimen adequacy, sensitivity and negative predictive value between left and right liver lobes are probably due to the greater difficulty in sampling focal liver lesions located in the right lobe for anatomical reasons, although technical success was the same for both lobes. Another strength of our study is the uniformity of the tissue acquisition method, since all the patients exclusively underwent fine needle biopsy using the same technique (wet suction), thus reducing the heterogeneity of the results. However, our study does have several limitations. First, this is a single-arm retrospective analysis that lacks a control arm. Second, there is a relatively small sample size, although the number of patients included is in line with other studies in the literature conducted on the EUS-FNB of focal liver lesions. Finally, the variability of the needles used in terms of gauge and design may impact diagnostic yield and the purity of the results.

## 5. Conclusions

Although these data are retrospective, EUS-FNB of focal liver lesions using the wet suction technique is effective in making an accurate diagnosis, with an excellent safety profile, for lesions located in both liver lobes. Our data suggest that the main indication for EUS-guided focal liver lesion biopsy is when obtaining a specimen from lesions located in the left liver lobe, close to big vessels, or in obese patients, where obtaining biopsies percutaneously is more challenging. However, prospective studies are needed to further define the best indications, the optimal set of patients who can benefit from this procedure, and the optimal technique and type of FNB needle to use.

## Figures and Tables

**Figure 1 diagnostics-14-01336-f001:**
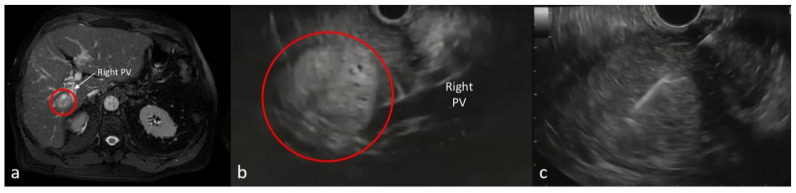
(**a**) Contrast-enhanced MRI showing a solid hyperenhancing lesion of the VI segment (red circle) close to the bifurcation of the right portal vein. (**b**) EUS view of the lesion in proximity to the right portal vein; (**c**) EUS view of the FNB needle in the lesion. Histopathologic diagnosis was consistent with metastasis from a neuroendocrine tumor.

**Table 1 diagnostics-14-01336-t001:** Characteristics of patients at baseline.

Characteristics of Patients at Baseline	Value
Patients *n*.	58
Male *n*. (%)	32 (55%)
Age (mean, SD)	61.5 ± 13.2
Caucasian *n*. (%)	58 (100%)
BMI at baseline (mean ± SD)	28 ± 2.7
Chronic liver disease (mean, %)	2 (3%)

**Table 2 diagnostics-14-01336-t002:** Characteristics of focal liver lesions and procedures.

Characteristics of Focal Liver Lesions	*n*. (%)
Right liver lobes	16 (27.6)
Left liver lobes	42 (72.4)
Transgastric route	42 (72.4)
Transduodenal route	16 (27.6)
Segment I	5 (8)
Segment III	11 (19)
Segment IV	14 (24)
Segment V	12 (20)
Segment VI	6 (11)
Segment VII	1 (2)
Segment VIII	1 (2)
Mean size in cm	34.3 ± 21.5

**Table 3 diagnostics-14-01336-t003:** Final histological diagnosis of focal liver lesions.

Histological Diagnosis	*n*. (%)
Malignancy	43 (74)
Hepatocellular carcinoma	7 (15)
Cholangiocarcinoma	16 (38)
Metastasis	16 (38)
Malignant neoplasm of unclear origin	3 (7)
Kaposi sarcoma	1 (2) 12 (20)
Benign disease	12 (21)
Specimen suboptimal	3 (5)

**Table 4 diagnostics-14-01336-t004:** Overall diagnostic performances of EUS-FNB.

Overall Diagnostic Performances of EUS-FNB	%
Sensitivity	95
Specificity	100
Adequacy	95
Accuracy	96

**Table 5 diagnostics-14-01336-t005:** Subgroup analysis for different gauge and type of needle used.

FNB Needle	*n*	Adequacy (%)	Accuracy (%)
19-gauge FNB needle	22	100	95
20-gauge FNB needle	6	100	100
22-gauge FNB needle	30	90	96
Acquire needle	52	94	95
ProCore needle	6	100	100

**Table 6 diagnostics-14-01336-t006:** Diagnostic yield of EUS-FNB between left and right liver lobes.

Diagnostic Yeld	Left Liver Lobe	Right Liver Lobe	*p*-Value
Sensitivity	96	91	0.001
Specificity	100	100	
Positive predictive value	100	100	
Negative predictive value	90	50	0.001
Adequacy	100	81	0.001

**Table 7 diagnostics-14-01336-t007:** Adverse events of EUS-FNB.

Adverse Events of EUS-FNB	*n*. (%)
Adverse events	0%

## Data Availability

The datasets used and/or analyzed during the current study are available from the corresponding author on reasonable request.

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
