# Peer review of "Endoscopic Ultrasound-Guided Fine Needle Biopsy of Focal Liver Lesions: An Effective Mini-Invasive Alternative to the Percutaneous Approach"

_diagnostics, 2024, doi:10.3390/diagnostics14131336_

Round 1
Reviewer 1 Report
Comments and Suggestions for Authors
This retrospective analysis of 58 patients undergoing EUS-guided fine needle biopsy of focal liver lesions shows excellent sensitivity and specificity. I have a few comments:
- The introduction is lengthy and contains topics better suited and already discussed in the discussion section.
- It should be mentioned that fine needle biopsies are also suitable for molecular analyses e.g. next generation sequencing.
- Overall, the message is rather simple and clear. Therefore, I recommend shortening the manuscript substantially.
The manuscript needs extensive language editing.
Author Response
- The introduction is lengthy and contains topics better suited and already discussed in the discussion section.
We agree with you. We have shortened the introduction.
- It should be mentioned that fine needle biopsies are also suitable for molecular analyses e.g. next generation sequencing.
We agree with you. Added to line 99-101.
- Overall, the message is rather simple and clear. Therefore, I recommend shortening the manuscript substantially.
We have shortened the manuscript.
Reviewer 2 Report
Comments and Suggestions for Authors
Overall, this study contributes valuable insights into the role of EUS-FNB as a minimally invasive alternative for liver biopsy. Congratulations to the authors.
Author Response
Thank you very much!
Reviewer 3 Report
Comments and Suggestions for Authors
The abstract is much too long.
The background part is too long and detailed. More than 20 studies were cited here but with no conclusion what issues are still not answered. After reading it I have a feeling that the subject is already well studied, and I see no point in the next study. The division into studies using different needles or considering whole liver or only left lobe should be done. The controversial issues should be underlined. What are the drawbacks of the previous studies?
What is special in this work to publish it? Even the first study from 1999 was bigger than this one.
Writing the generally known formulas for sensitivity, PPV and NPV seems unnecessary.
What did the choice of the needle size and type depend on?
It should be underlined that only 2 patients had the lesions in most distant from the stomach and duodenum segments (7,8). In which segments were the lesions for which the biopsy was not diagnostic?
What were the benign diagnoses?
Minor remarques:
Lines 36-38 – the sentence sounds awkward. Lack of sedation should not be the first argument because in theory percutaneous biopsy may be performed in sedation too.
Line 40-41 – why biopsy under the guidance of USG is not real time guided?
Lined 46-47
Is trans jugular biopsy done also in focal lesions or only in diffuse liver diseases? If it is, how is it guided?
Line 78 – contrast enhancement.
Comments on the Quality of English LanguageMinor improvements needed.
Author Response
The answers are in the attached file.

Round 2
Reviewer 1 Report
Comments and Suggestions for Authors
The revised manuscript is substantially improved. The key messages are clearly presented. The discussion is still lengthy and could be shortened by focusing on the data collected.
Comments on the Quality of English LanguageMinor improvements in languge are needed.
Author Response
Thanks for the comment. We have further reduced the discussion by eliminating the most discursive part. Now, we focused on the introduction to the discussion, on the data collected in the literature and on the strengths and weaknesses of the study.
Reviewer 3 Report
Comments and Suggestions for Authors
The article is fine now. The only comments to the beginning of discusion- you should not write that percutaneous biopsy does not need special training.
Comments on the Quality of English LanguageMinor corrections needed.
Author Response
Thanks for the comment.
Rereading it, we agree with you and it was deleted.